RESEARCH CULTURE

# Co-reviewing and ghostwriting by early-career researchers in the peer review of manuscripts

**Abstract** Many early-career researchers are involved in the peer review of manuscripts for scientific journals, typically under the guidance of or jointly with their advisor, but most of the evidence about this activity is anecdotal. Here we report the results of a literature review and a survey of researchers, with an emphasis on co-reviewing and 'ghostwriting'. The literature review identified 36 articles that addressed the involvement of early-career researchers in peer review, most of them about early-career researchers and their advisors co-reviewing manuscripts for the purposes of training: none of them addressed the topic of ghostwriting in detail. About three quarters of the respondents to the survey had co-reviewed a manuscript. Most respondents believe co-reviewing to be a beneficial (95%) and ethical (73%) form of training in peer review. About half of the respondents have ghostwritten a peer review report, despite 81% responding that ghostwriting is unethical and 82% agreeing that identifying co-reviewers to the journal is valuable. Peer review would benefit from changes in both journal policies and lab practices that encourage mentored co-review and discourage ghostwriting.
DOI: https://doi.org/10.7554/eLife.48425.001

**GARY S MCDOWELL\*, JOHN D KNUTSEN, JUNE M GRAHAM, SARAH K OELKER AND REBECCAH S LIJEK\***

**\*For correspondence:**
garymcdow@gmail.com (GSMD);
rlijek@mtholyoke.edu (RSL)

## Introduction

The peer review of manuscripts submitted to scientific journals is widely viewed as fundamental to efforts to maintain the integrity of the scientific literature (*Baldwin, 2018*; *Tennant, 2017*). Early-career researchers (ECRs) often contribute to the peer review process. Indeed, in a recent survey that targeted ECRs in the life sciences, 92% of respondents reported that they had been involved in the peer review of at least one manuscript (*Inside eLife, 2018*). More than half of survey respondents, including 37% of graduate students, reported reviewing a manuscript without any assistance from their advisor. Journals may not be fully aware of the extent to which ECRs are involved in peer review (*McDowell, 2018*). Indeed, a recent editorial in this journal contained the following sentence: "It is common practice for busy group leaders to ask their more senior PhD students and postdoctoral fellows to help with peer review, but in too many cases these contributions go unacknowledged" (*Patterson and Schekman, 2018*).

We conducted a literature search and a survey to explore the involvement of ECRs in peer review more thoroughly and, in particular, to determine the prevalence of both co-reviewing (i.e., when the journal knows that the ECR contributed to the review) and 'ghostwriting' (i.e., when the journal does not know that the ECR contributed to the review). Please see *Table 1* for a definition of terms used in this article.

## Results

### Lack of literature on ECR ghostwriting of peer review reports

We performed a comprehensive review of the peer-reviewed literature to identify any previous studies on the role that ECRs play in peer review, particularly with respect to ghostwriting. Exhaustive search terms that combined any

synonyms of "early-career researcher" and "peer review" were used per evidenced-based guidelines for systematic reviews (prisma-statement.org; *PRISMA Group et al., 2009*; see Methods: Systematic literature review for details). Our search yielded 1952 unique articles. Collected articles underwent two rounds of screening performed independently by three of the present authors using titles and abstracts to evaluate relevance to the topic of ECR co-reviewing and ghostwriting peer review reports (see Methods: Relevance screening for details; *Supplementary file 1*; *Figure 1—figure supplement 1*).

We did not find any research articles on ghostwriting peer review reports by ECRs. One article (the eLife editorial mentioned previously; *Patterson and Schekman, 2018*) acknowledged the phenomenon of ECR ghostwriting, and stated that ECRs are eligible to act as peer reviewers for manuscripts submitted to eLife. 35 other articles addressed ECR involvement in manuscript peer review but did not address ghostwriting. Many of these instead investigated the value of co-reviewing as a training exercise (see Appendix 1). None discussed the issue of named credit for scholarly labor, nor did they include information on the frequency of ghostwriting in peer review or the opinions of ECRs on ghostwriting.

## Surveying the rates and rationales for co-reviewing and ghostwriting

To address this gap in the literature, we designed a survey to evaluate the frequency of, and motivations for, ghostwriting and co-reviewing by ECRs. The IRB-approved, online survey garnered 498 responses over a month-long data collection period in September, 2018 (see Methods: Survey of peer review experiences and attitudes for details; the survey itself is available in *Supplementary file 2*). Respondents came from 214 institutions that were geographically diverse both within and beyond the United States. Most participants were from institutions in North America (n = 370), Europe (n = 87) and Asia (n = 21). 74% of all respondents were based in the US, of which 64% were citizens or permanent residents, and 36% held temporary visitor status. Institutions from 40 US states or territories were represented: the four universities with the most respondents were Washington University in St. Louis, the University of Kentucky, Rockefeller University, and the University of Chicago. The majority of survey respondents (65%) were ECRs in the life sciences (*Figure 1*). This was as expected given our efforts to primarily engage ECRs and our connections to biomedical post-doctoral populations (see Methods: Survey distribution, limitations, and future directions). We surmise that postdocs (63% of all respondents) are over-represented in this survey, although the

**Table 1.** Definitions used in this study.

| Term | Definition |
| --- | --- |
| Early-career researcher (ECR) | We consider this to be anyone engaged in research who is not recognized as an independent leader of a research group, including: undergraduate, graduate, and postdoctoral researchers; junior research assistants. |
| Principal Investigator (PI) | Anyone recognized as an independent leader of a research group, including: professors, group leaders.<br>*Note*: We use this term to mean someone likely to be an invited reviewer due to their professional independence, including pre-tenure junior faculty (e.g. assistant professor in the US). We recognize that, in other contexts, pre-tenure faculty may also categorized as ECRs. |
| Co-reviewing | Contributing ideas and/or text to a peer review report when one is not the invited reviewer. Equivalent to a co-author on a manuscript when one is not the corresponding author.<br>*Note*: We use this term to mean significant contributions to the peer review report, and so differentiate from casual or insignificant conversations about the manuscript under review that do not provide novel ideas and/or text to the peer review report. |
| Ghostwriting | Co-reviewing without named credit to the journal editorial staff.<br>*Note*: We use this term to mean only the identification of a co-reviewer to the journal staff in an identical manner to the identification and naming of the invited reviewers. We are not referring to the *public* naming of peer reviewers, or reviewers *signing* reviews, or other forms of open peer review which is beyond the scope of this study (*Ross-Hellauer, 2017*). |

DOI: https://doi.org/10.7554/eLife.48425.002

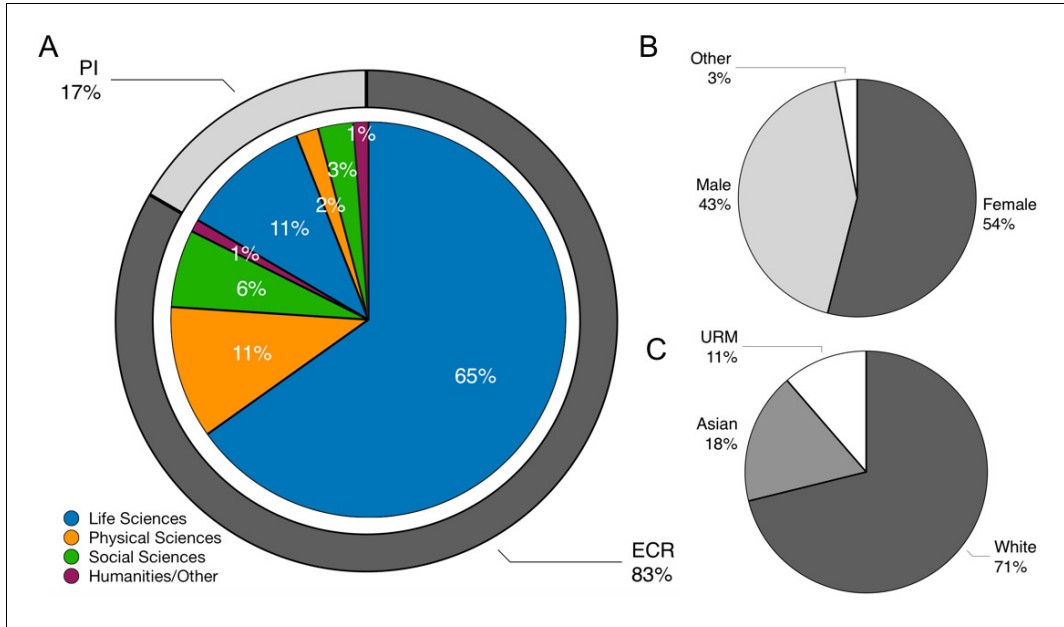

**Figure 1.** Demographics of survey respondents. (**A**) Distribution of responses by field of study and career stage. Of a total of 498 respondents, 488 were categorized as an early career researcher (ECRs; n = 407/488; 83%) or principal investigator (PIs; n = 81/488; 17%). Of these, 76% were in the life sciences (318 ECRs; 52 PIs), 13% were in the physical sciences (53 ECRs; 9 PIs), 9% were in the social sciences (31 ECRs; 14 PIs), and 2% were in the humanities/other (5 ECRs; 6 PIs). 10 respondents were neither ECR nor PI (e.g., "unemployed"; data not shown). (**B**) Distribution of responses by gender: 54% (271/498) of respondents were female, 43% (216/498) were male, and 3% (11/498) provided another or no response. (**C**) Distribution of responses by race/ethnicity: Of the 481 respondents who provided an answer to this question, 71% (342/481) were coded as white, 18% (84/481) Asian, and 11% (55/481) URM (underrepresented minority in the sciences).

DOI: https://doi.org/10.7554/eLife.48425.003

The following source data and figure supplement are available for figure 1:

**Source data 1.** De-identified demographic data for survey respondents.
DOI: https://doi.org/10.7554/eLife.48425.005

**Figure supplement 1.** Search strategy for literature review with number of records remaining at each stage.
DOI: https://doi.org/10.7554/eLife.48425.004

number of postdoctoral researchers in the US is currently unknown (*Pickett et al., 2017*).

### Co reviewing by ECRs is the norm and motivated by training

It is a widespread practice to contribute ideas and/or text to a peer review report when one is not the invited reviewer. 73% of all respondents have co-reviewed and often at numerous times (33% on 6–20 occasions, and 4% on more than 20 occasions; *Figure 2A*). Co-reviewing by ECRs specifically is common, with 79% of postdocs and 57% of PhD students having co-reviewed when "the invited reviewer is the PI for whom you work" (*Figure 2B*). These data suggest that collaboration on peer review reports is an academic norm, especially by ECRs who are not the invited reviewer. By contrast, 55% of ECR

respondents have never carried out independent peer review as the invited reviewer (*Figure 2C*).

A major motivation for ECRs to co-review is to gain training in peer review of manuscripts, a fundamental scholarly skill. All survey respondents were asked what training they received in peer review of manuscripts (*Figure 3*). Respondents report that PIs are the second most common source of peer review training, bested only by the passive form of learning "from receiving reviews on my own papers." Training through co-review was the subject of many publications uncovered by our literature review (Appendix 1).

### Ghostwriting is common despite a belief it is unethical

The frequency of ghostwriting was measured in two survey questions which revealed comparable rates. When we asked "To your knowledge,

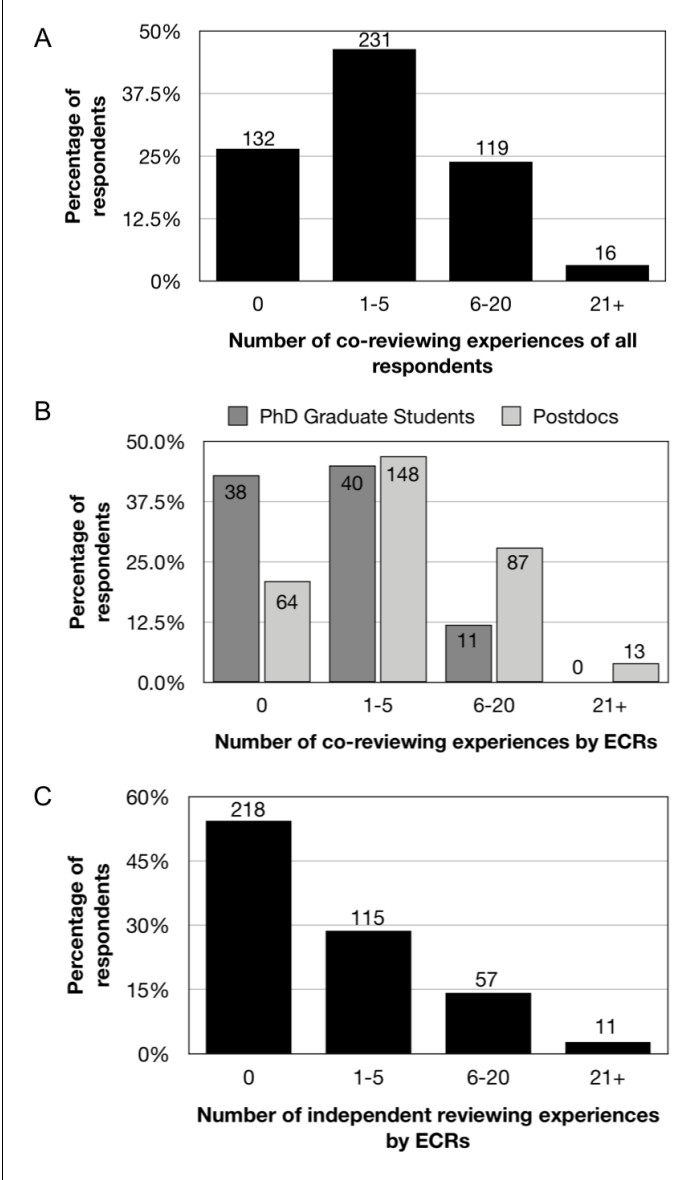

**Figure 2.** Experiences of co-reviewing and being invited to review. (A,B) Responses to question: "How many times in your career have you contributed ideas and/or text to peer review reports where you are not the invited reviewer (e.g. the invited reviewer is the PI for whom you work)?" 73% of all respondents (366/498) had participated in co-reviewing: 63% of this subsample had carried out co-reviewing activities on 1–5 occasions; 33% on 6–20 occasions; and 4% on more than 20 occasions. (B) Number of co-reviewing experiences by career stage for 401 ECRs: the distribution of postdocs (n = 312) is skewed toward more co-reviewing experiences, whereas the distribution of PhD students (n = 89) is skewed toward fewer co-reviewing experiences. (C) Responses to question for ECRs: "How many times in your career have you reviewed an article for publication independently, i.e. carried out the full review and been identified to the editorial staff as the sole reviewer?" 55% of the ECR respondents (218/401) had never carried out independent peer review, and 46% (183/401) had carried out independent review as the invited reviewer: 115 had done so 1–5 times, 57 had done so 6–20 times, and 11 had done so more than 20 times.

DOI: https://doi.org/10.7554/eLife.48425.006

did your PI ever withhold your name from the editorial staff when you served as the reviewer or co-reviewer?," 46% of respondents knew that their name had been withheld (*Figure 4A*). These data are consistent with results from a separate question: "When you were not the invited reviewer, what was the extent of your involvement in the peer review process?". For this question, 44% of respondents reported having had the experience of ghostwriting: "I read the manuscript, wrote the report, my PI edited the report and my PI submitted report with only their name provided to the editorial staff" (*Table 2*). Taken together, these data suggest that approximately 1 in 2 survey respondents has engaged in ghostwriting of a peer review report on behalf of their PI, the invited reviewer. Furthermore, 70% of co-reviewers report the experience of making significant contributions to a peer review report without knowingly receiving credit (*Table 2*). This experience is much more common than the 22% of co-reviewers who experienced making significant contributions with known credit (*Table 2*). These data reveal a breakdown in communication between invited reviewers and co-reviewers.

In a more specific follow up question that asked "To your knowledge, did your PI ever submit your reviews without editing your work?", 52% of respondents report that they were not involved in any editing process with their PI (*Figure 4B*). This proportion is similar to that reported in *Inside eLife (2018)*. That survey asked "Have you reviewed before?" and then "If so, to what extent was your supervisor involved?" to which slightly more than half of the 264 respondents replied "not at all." One interpretation of these data is that half of respondents had engaged in independent peer review as the invited reviewer. Another interpretation of these data is that half of respondents had engaged in co-review with no feedback from their supervisor, the invited reviewer. Our data support the latter interpretation that slightly more than half of respondents have written peer review reports without feedback from their PI when the PI is the invited reviewer.

The Office of Research Integrity (ORI), which oversees research funded by the US Public Health Service, states that "academic or professional ghost authorship in the sciences is ethically unacceptable" (https://ori.hhs.gov/plagiarism-34). Respondents are also of the view that ghostwriting peer review reports is unethical: 83% disagree with the statement that "The only person who should be named on a peer

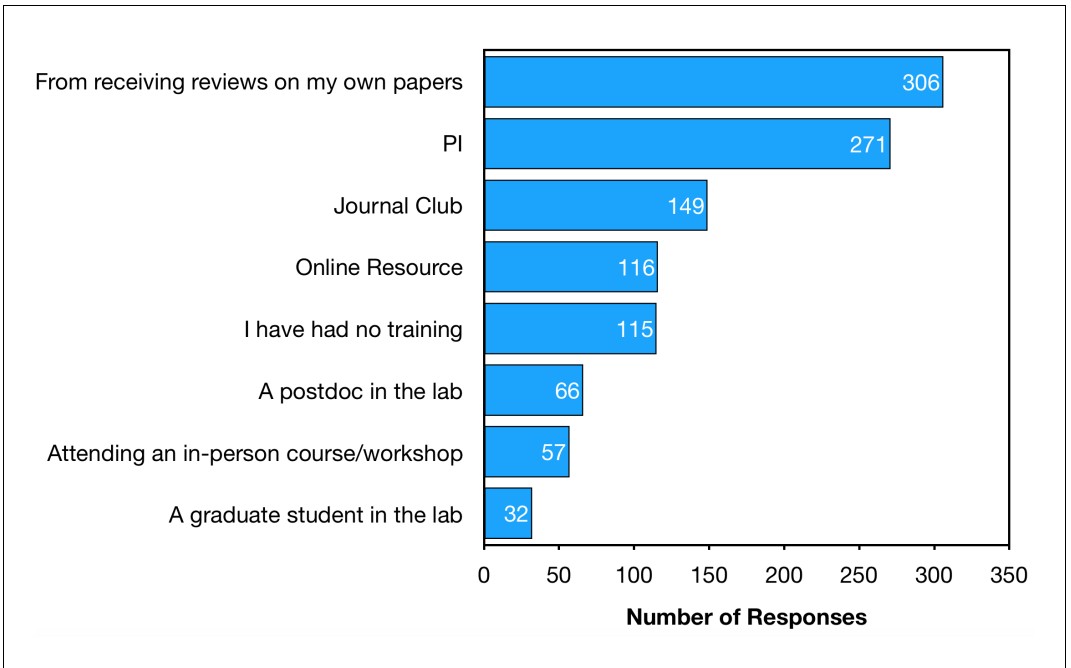

**Figure 3.** Training in how to peer review a manuscript. Responses to the question: "How did you gain training in how to peer review a manuscript?" Respondents were able to select as many options as applied to them. These data include responses from all survey participants, including those without any independent or co-reviewing experience.
DOI: https://doi.org/10.7554/eLife.48425.007

review report is the invited reviewer, regardless of who carried out the review"; 81% disagree with the statement that "Ghostwriting a peer-review report for your PI is an ethically sound scientific practice"; and 77% disagree with the statement that "It is ethical for the invited reviewer (e.g. PI) to submit a peer review report to an editor without providing the names of all individuals who have contributed ideas and/or text to the report" (*Figure 5*). Respondents were also supportive of co-reviewing providing their contributions are known about: 74% agree that "Anyone that contributes ideas and/or text to the review report should be included as a co-author on the review"; 82% agree that "It would be valuable to have my name added to a peer review report (e.g. to be recognized as a co-reviewer by the editor; or to use a service such as Publons to be assigned credit)"; 73% agree that "It is ethical for the invited reviewer (e.g. PI) to involve others (e.g. their trainees) in reviewing manuscripts"; and 95% agree that "Involving members of a research group in peer review is a beneficial training exercise." The latter statement evoked the strongest positive sentiment of all 11 Agree/Disagree statements.

There was a significant difference in the extent of agreement between ECRs and PIs for certain aspects of co-reviewing and ghostwriting (*Table 3*). In 3 of 11 statements, ECRs felt significantly more strongly than PIs but still shared the same valance (e.g. both groups agreed or both groups disagreed with the statement, just to a differing strength). For the remaining 8 statements, ECRs and PIs did not significantly differ in their opinions.

### Motivations for ghostwriting

If 4 out of 5 survey respondents think ghostwriting is unethical, then why do half of all respondents participate? We measured the motivations for ghostwriting by: i) asking all respondents, regardless of peer review experience, to surmise why someone might withhold the name of a co-reviewer (*Figure 6*); ii) asking only respondents with ghostwriting experience to report the specific reasons that the invited reviewer gave for withholding their name (*Table 4*). In this way, we compared cultural beliefs with actual practice.

The main perceived barrier to naming co-reviewers was a lack of a physical mechanism to supply the name to the journal (e.g. a textbox for co-review names), with 73% of respondents

**Table 2.** Experiences with co-review and ghostwriting.

Responses to the question: "When you were not the invited reviewer, what was the extent of your involvement in the peer review process?" Survey participants were able to choose any and all applicable responses from a provided set of possible responses that can be broken down into three interpretation groups. Because respondents were able to select more than one answer, these data include all of the different co-reviewing experiences for each participant.

| Possible survey responses | Respondents that selected this as an answer (%) | Interpretation of response | Respondents that selected at least one of the answers in this group (n, %) |
|---|---|---|---|
| "I read the manuscript, shared short comments with my PI, and was no longer involved" | 40 | No significant contribution | 149 respondents (**40%**) selected this response |
| "I read the manuscript, wrote a full report for my PI, and was no longer involved" | 47 | Significant contribution, <u>without</u> known credit | 258 respondents (**70%** of those with co-reviewing experience) selected at least one of the responses in this category |
| "I read the manuscript, wrote the report, my PI edited the report and my PI submitted report with only their name provided to the editorial staff" | 44 | | |
| "I read the manuscript, wrote the report, my PI edited the report and we submitted the report together with both of our names provided to the editorial staff" | 20 | Significant contribution, <u>with</u> known credit | 80 respondents (**22%** of those with co-reviewing experience) selected at least one of the responses in this category |
| "I read the manuscript, wrote the report, and submitted it independently without my PI's name provided to the editorial staff" | 3 | | |

Note: (Mis)representation of authorship on any scholarly work can be a subjective grey area. We sought to specifically avoid this in our survey questions by using the answers to the question "When you were not the invited reviewer, what was the extent of your involvement in the peer review process?" to disambiguate the grey areas of authorship. We consider any experience that began with "I read the manuscript, wrote a full report for my PI, and…" to be an unequivocally significant contribution deserving of authorship on the peer review report.

DOI: https://doi.org/10.7554/eLife.48425.009

selecting this as an option (*Figure 6*). Cultural expectations were the next most commonly-cited barriers, such as "A belief that reviews should only be done by the invited reviewer, and not by, or with assistance from, anyone else" (selected by 63% of respondents) and "A belief that including co-author information would demonstrate that the PI breached the confidentiality of the manuscript" (58%). These latter responses allude to journal policies prohibiting invited reviewers from sharing unpublished manuscripts without prior permission. Write-in responses echo themes about how ghostwriting is the status quo in peer review (*Table 5*). At the same time, respondents also wondered why including co-reviewer names is not common practice.

In contrast, when we asked "Consider cases where you contributed to a peer review report and you know your name was NOT provided to the editorial staff. When discussing this with your PI, what reason did they give to exclude you as a co-reviewer?" 73% of respondents reported that they had not discussed this with their PI (*Table 4*). This is consistent with the lack of communication between invited reviewers and co-reviewers documented above (*Table 2*; *Figure 4B*). Of the 27% of respondents who had ghostwritten and did discuss the matter with

their PI, most were told that the reason their name was withheld was either a prohibitive journal policy and/or prevailing cultural expectations about co-authorship on peer review reports. Only 4% of those who had discussed the matter with their PI cited a practical barrier, such as the lack of a text box for co-reviewer names on the journal review submission form. Write-in responses to this question again refer to cultural expectations as the major drivers for ghostwriting (*Table 6*). Many of these write-in reasons articulate that it is good practice for ECRs to participate in peer review; however, they simultaneously fail to explain why this necessitates withholding the names of co-reviewers. These data suggest that there is a common conflation of ghostwriting (withholding names) with co-reviewing (involving ECRs in peer review, often for the purposes of training).

### Other demographic analyses

We also performed preliminary analyses by gender, field, citizenship and race/ethnicity, but any differences we observed were small or not statistically significant, and some demographic subsets were too small for analysis. We are considering how to share these data more completely.

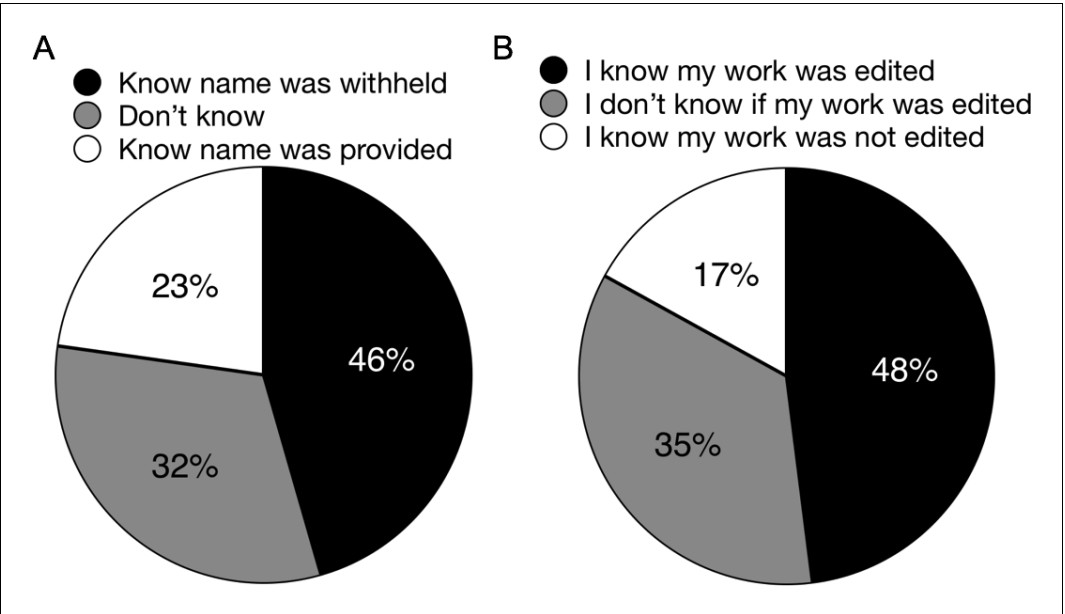

**Figure 4.** The actions of PIs during co-review. (**A**) Responses to the question: "To your knowledge, did your PI ever withhold your name from the editorial staff when you served as the reviewer or co-reviewer?" 46% of respondents (171/374) knew that their name had been withheld, and 32% (118/374) did not know. The remaining 23% (85/374) responded that they knew for certain their name had been disclosed. (**B**) Responses to the question: "To your knowledge, did your PI ever submit your reviews without editing your work?" 17% of respondents (66/375) answered "yes", that they knew that their work had not been edited by the PI prior to submission to the journal. Another 35% of respondents (132/375) were unaware of whether their work was edited by their PI prior to their PI submitting it to the journal. Taken together, these 52% of respondents were not involved in editing, regardless of whether it took place. 48% of respondents (177/375) answered "no", indicating that they knew their work had been edited for sure.

DOI: https://doi.org/10.7554/eLife.48425.008

## Discussion

Journal peer review is an important part of scholarship. As such, ECR training and authorship on peer review reports deserves thoughtful consideration to ensure the integrity of the peer review process. Our data reveal that involving ECRs in peer review is an academic norm, with about three-quarters of our survey population having contributed significantly to a peer review report when they were not the invited reviewer (co-reviewed), and about half having done so without being named to the journal editorial staff (ghostwritten). These high frequencies contrast with journal policies and cultural expectations that only the invited reviewer engages in the peer review of a manuscript. They also fly in the face of community values when about four-fifths of those surveyed agree that ghostwriting is unethical. What drives these differences between community values and experience?

### Explanations for ghostwriting are conflated with explanations for co-reviewing

Co-reviewing as a training exercise and ghostwriting are separable processes: training through co-review can and does happen *whether or not* named credit is given to the co-reviewer, and excluding co-reviewer names from peer review reports can and does happen *whether or not* the co-reviewer has experienced quality training in the process. Even as we sought to collect data that would disentangle these two processes, the rationales for ghostwriting were often conflated with the rationales for co-reviewing. For example, when we asked respondents specifically for the reason(s) their PI gave them for excluding their names on a peer review report, many wrote responses such as "I was told this is how one gets to train to review papers..." (*Table 6*). This response – that it is a beneficial and common practice for ECRs to participate in peer review as a training exercise – does not actually explain why *ghostwriting*

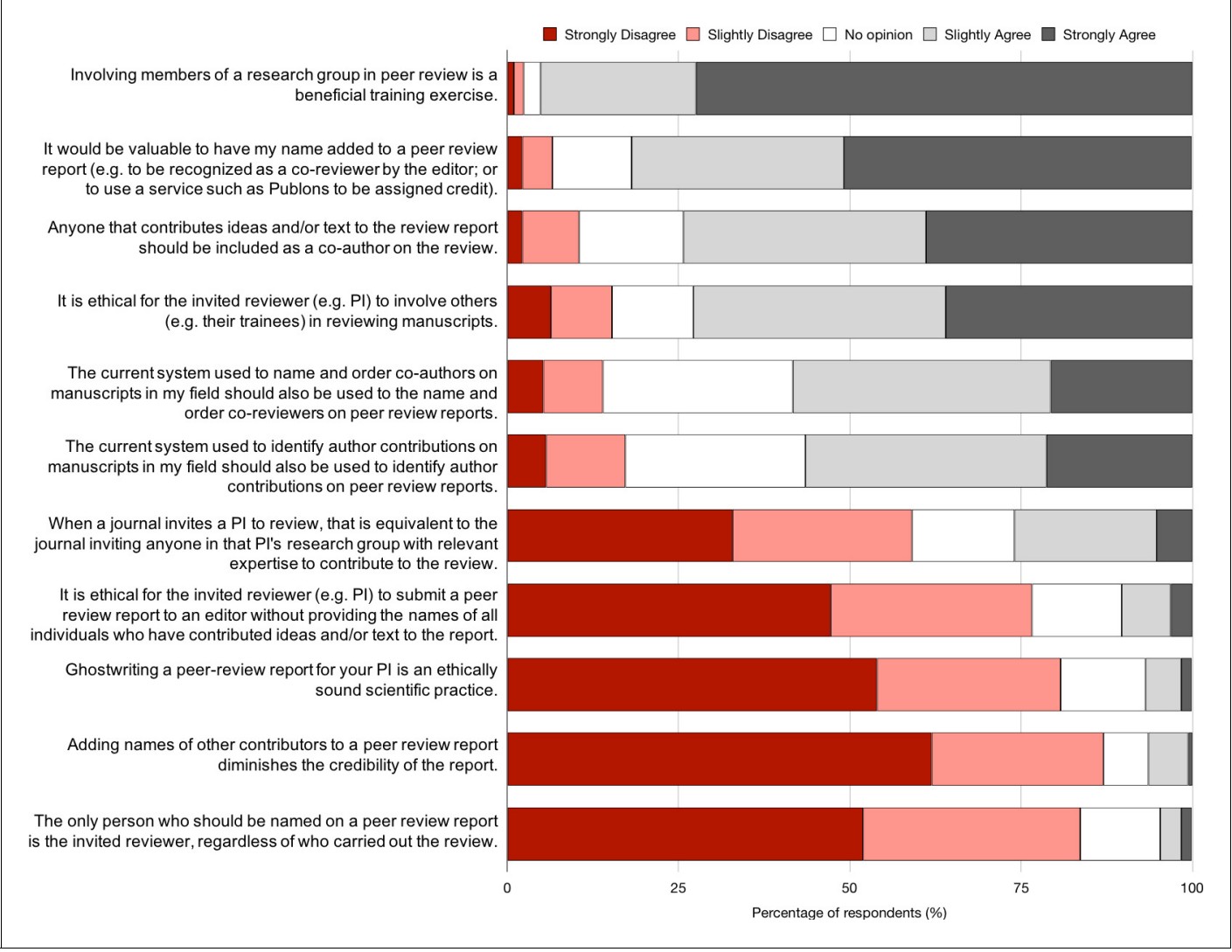

**Figure 5.** Views on co-review, ghostwriting, and other aspects of peer review. Responses to the question: "Please indicate how strongly you agree with the following statements." Data represent the opinions (not experiences) of all respondents regardless of whether or not they had participated in peer review. Respondents were also provided with a textbox to submit comments to expand and/or clarify their opinions.

DOI: https://doi.org/10.7554/eLife.48425.010

The following source data is available for figure 5:

**Source data 5.** Opinions on co-review and ghostwriting.

DOI: https://doi.org/10.7554/eLife.48425.011

occurs. Reducing ghostwriting requires decoupling it in the zeitgeist from the beneficial training of ECRs through co-review. We therefore separate our discussion of the motivations for co-reviewing and ghostwriting in an effort to find solutions to ghostwriting that simultaneously support ECR co-reviewing as training in peer review.

### Co reviewing by ECRs as valued training or delegation of scholarly labor

Survey respondents clearly find that co-reviewing by ECRs has significant benefits and is not inherently problematic when the issue of named credit is set aside. Co-review is considered an ethical (73% agree) and beneficial training exercise (95% agree), explaining why co-reviewing is the second most commonly reported source of training in peer review. These data should be weighed heavily when considering journal policies, since any policy that prevents co-reviewing

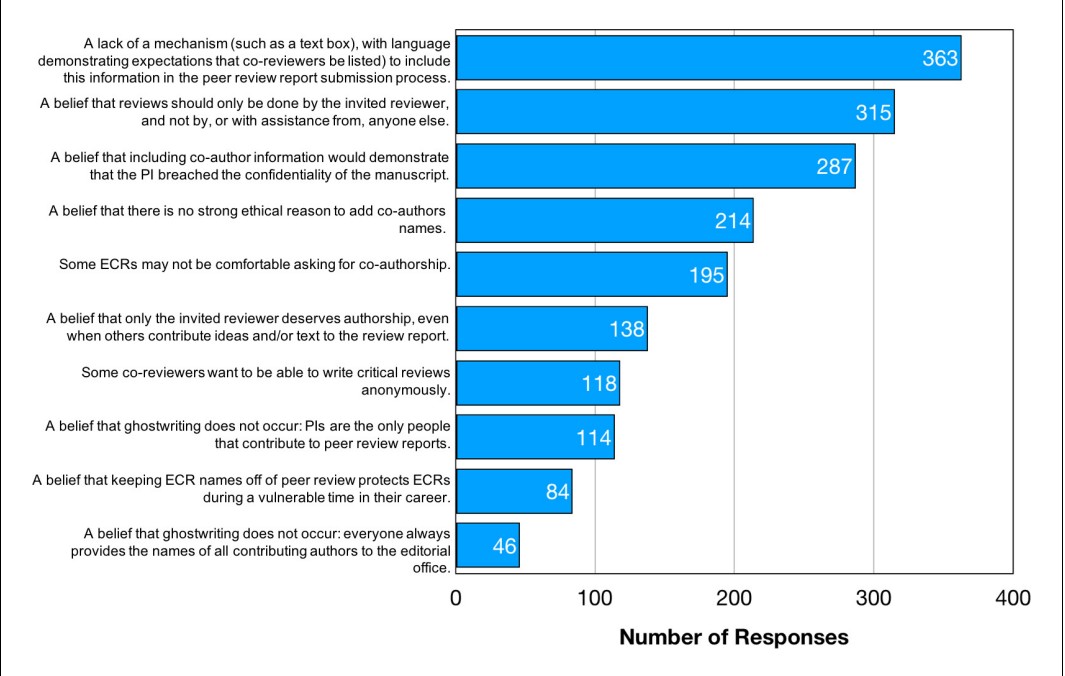

**Figure 6.** Reasons why journals might not know about co-reviewers. Responses to the question: "What do you think are the reasons why the names of co-authors on peer review reports may not be provided to the editorial staff?." Here our intent was to ask the respondents about the barriers that might cause names to be withheld (rather than asking whether they thought co-reviewers should be named). Respondents were able to select as many answers as they felt applied. The frequencies do not allow us to assess how important the barriers are, and respondents were not asked to rank barriers, but simply to surmise which ones they felt were relevant to the current practice of ghostwriting.

DOI: https://doi.org/10.7554/eLife.48425.013

by ECRs would remove a common and valuable training exercise in peer review. We support the adoption of policies that specifically embrace ECR co-review as training (e.g., eLife, 2019). The Transpose project is also compiling a crowd-sourced database of journal policies on peer review, co-reviewing and preprinting (*Transpose, 2019*).

"ECR training" is not a sufficient justification for co-reviewing for the half of survey respondents who have written a peer review report without any interaction with their PIs. In these cases, co-reviewing devolves into a delegation of scholarly labor that benefits the invited reviewer, often an overburdened PI in a hypercompetitive research environment (*Alberts et al., 2014*). This environment incentivizes the use of ECRs as cheap labor to fulfill productivity requirements, especially as the growth of the ECR population outpaces the growth of independent academic positions to employ them (*Heggeness et al., 2017*; *Heggeness et al., 2016*). These market forces provide a second explanation, beyond ECR training, for why co-reviewing is

commonplace. Our survey respondents agree, sharing sentiments like "apparently this duty is part of my job description" (*Table 6*). The delegation of scholarly labor to ECRs is not necessarily, nor intentionally, exploitative, although it can easily become so given the power dynamics and documented lack of communication between mentors and mentees (*Van Noorden, 2018*). Any successful intervention to address concerns about the ethics of co-reviewing by ECRs must take into account that it is commonly an unstated expectation that ECRs carry out peer review on behalf of their PI, and that ECRs may not feel they have the freedom to decline even if they feel it is unethical to participate.

### Limitations to depending on co-reviewing as training

Survey data suggest that training in peer review is determined by a small number of individual experiences outside of evidence-based training structures and community oversight. "Receiving reviews on my own papers" only gives a limited number of examples of how others review and is

**Table 3.** Statements for which the differences in the responses of the ECR and PI populations were statistically significant.
We calculated the mean degree of agreement by setting 1 as Strongly Agree through to 5 as Strongly Disagree, and 3 set as No Opinion. The higher the mean value calculated for the group, the closer the group feels to disagreeing with statement. "No opinion" responses, coded as 3, are included in these means. A 2-tailed student's t-test for equality of the means was used to calculate p values. Due to the difference in the percentage of ECRs and PIs with "no opinion" for the third question, we removed "no opinion" responses and recalculated the mean scores: the difference between the mean scores was reduced but remained significant (ECRs: $1.57 \pm 0.05$ (n = 365); PIs: $1.88 \pm 0.15$ (n = 64); p=0.048).

| Statement | ECR Mean Score | PI Mean Score | p value | % ECRs with no opinion | % PIs with no opinion |
|---|---|---|---|---|---|
| Involving members of a research group in peer review is a beneficial training exercise. | $1.32 \pm 0.03$ (n = 405) | $1.54 \pm 0.10$ (n = 81) | p=0.033 | 2.5 | 2.5 |
| It is ethical for the invited reviewer (e.g. PI) to involve others (e.g. their trainees) in reviewing manuscripts.* | $2.06 \pm 0.06$ (n = 406) | $2.37 \pm 0.14$ (n = 81) | p=0.029 | 11 | 15 |
| It would be valuable to have my name added to a peer review report (e.g. to be recognized as a co-reviewer by the editor; or to use a service such as Publons to be assigned credit). | $1.71 \pm 0.05$ (n = 405) | $2.11 \pm 0.13$ (n = 81) | p=0.003 | 10 | 21 |

*Indicates that p value was calculated assuming equal variance according to Levene's test for Equality of Variances.
DOI: https://doi.org/10.7554/eLife.48425.012

a passive form of learning that lacks individualized or iterative feedback from a mentor. Since it is a common complaint that reviews are overly critical (*Schneiderhan, 2013*), it seems counterintuitive for this to be the main example by which ECRs learn how to review. Training provided by one's PI may benefit from a personalized teaching relationship but depends on the PI's own training. The resulting trickle-down training is likely to be self-reinforcing and highly variable in quality and content. The small sample size of these different personal experiences may also reinforce bias, including selection bias where one's few experiences may not be representative of the population, memory biases where negative experiences (e.g. from receiving harsh reviews) are more readily remembered and so taught (*Kensinger, 2007*), and gender or other demographic biases currently being studied in the peer review process (*Murray et al., 2018*).

These results highlight an area of opportunity to improve and standardize training in peer review of manuscripts as a critical scholarly skill (Appendix 1). Since the top two reported forms of training involve PIs either as manuscript reviewers or ECR mentors, interventions to ensure PIs have received training in peer review and in communicating this skill to ECRs may also be appropriate. A lack of "training the trainers" was cited as a main reason for why pairing experts with new peer reviewers failed to improve review quality in one of the few randomized controlled trials of this practice (*Houry et al., 2012*). Given their key role as the main source of training in peer review, it is

important that PIs make deliberate efforts to teach their trainees this skill and to provide their trainees with feedback. Mentorship from PIs may be complemented with, but not replaced by, journal clubs where ECRs gain experience in reviewing published manuscripts or preprints (see, for example *Avasthi et al., 2018* and *PRE-review, 2019*). Since these ad hoc reviewing experiences with PIs and journal clubs may vary in their availability and quality, all ECRs should be offered standardized, evidence-based training in peer review. For example, peer review courses that are compulsory and ubiquitous in graduate schools would ensure that all PhD-holders are appropriately trained to perform constructive peer review.

### A lack of communication about authorship of peer review reports

Even in the best case scenario for co-reviewing, when training is taking place and is effective, its benefits can still be confounded by ethical lapses such as ghostwriting. The most common explanation for ghostwriting was that authorship was simply not discussed (*Table 4*; 73% of responses from those who knew their names had been withheld). 47% of respondents have had the experience of "I read the manuscript, wrote a full report for my PI, and was no longer involved" (*Table 2*), a concerning breakdown in communication between invited reviewers and those actually writing the peer review report. At best, writing a peer review report without receiving feedback from one's PI is a missed opportunity for training. At worst, a peer review report that is written by one person and submitted to

**Table 4.** Reasons given by PIs for not naming co-reviewers to the journal editor.
Responses to the question: "Consider cases where you contributed to a peer review report and you know your name was NOT provided to the editorial staff. When discussing this with your PI, what reason did they give to exclude you as a co-reviewer?" In addition to the possible answers provided by the survey, respondents were also provided with a textbox to add write-in responses.

| Reasons given by PIs for not naming co-reviewers | Number of Respondents |
|---|---|
| Did not discuss with my PI | 210 |
| Co-authorship is for papers, not for peer review reports; Intellectual contribution not deemed sufficient | 33 |
| Journal requires prior approval to share manuscript, which was not obtained; Journal does not allow ECRs to review | 30 |
| Write-in answers citing mechanistic barriers (e.g. lack of a text box to enter co-reviewer names) | 3 |

DOI: https://doi.org/10.7554/eLife.48425.014

the journal under the name of another person is a breach of academic integrity.

Power imbalances may prevent an ECR from feeling able or willing to initiate this conversation with their PI. 39% of respondents think that ghostwriting occurs because "some ECRs may not be comfortable asking for co-authorship" (*Figure 6*). As one write-in states: "you don't want to piss off the boss." Ghostwriting, therefore, may be a symptom of the larger problem in academia that ECRs are extremely dependent on the good will of their PI for retention in the hypercompetitive research environment (for example, for letters of recommendation throughout their career or immigration status). Another reason that authorship may not be discussed is that not naming co-reviewers is the expected status quo: "[PI] didn't think of including me" and "they forgot" (*Table 6*). One postdoc added: "I'd never really thought about this before. I just assumed it was part of the process. But it is very time consuming and I do believe that all reviewers should receive credit for the review." If ghostwriting arises from PIs and ECRs simply not thinking to include co-reviewer names vs. intentional withholding of names, then building awareness should encourage more conversation about this issue and better mentoring practices will help overcome such miscommunication. PIs could be further incentivized to name ECR co-reviewers by accounting for this practice when they are evaluated for grant funding or tenure. For example, perhaps PIs who pay the salaries of their trainees from grant budgets should demonstrate in grant progress reports how those ECRs are being trained, which might include a list of which trainees were listed as co-reviewers.

### An expectation that co-reviewers do not deserve credit regardless of what they contribute

Ghostwriting is also driven by a cultural expectation that co-reviewers do not deserve named credit to the journal regardless of how much they contribute. 43% of ghostwriting experiences were explained by "co-authorship is for papers, not for peer review reports" or "intellectual contribution not deemed sufficient." 28% of respondents surmise that ghostwriting occurs due to "a belief that only the invited reviewer deserves authorship, even when others contribute ideas and/or text to the review report." These rationales for ghostwriting contradict community opinion about whether this should be the case, with 83% disagreeing that "the only person who should be named on a peer review report is the invited reviewer, regardless of who carried out the review" and 74% agreeing that "anyone that contributes ideas and/or text to the review report should be included as a co-author on the review." Ghostwriting could therefore be reduced if cultural expectations were shifted to reflect consensus opinion that co-reviewers deserve to be named to the journal.

### Prohibitive journal policy is out of alignment with current practice

Many journals have policies that prevent invited reviewers from sharing manuscripts with anyone else and/or policies that prevent ECRs from serving as reviewers or co-reviewers without prior permission from the editor (*Transpose, 2019*). These policies are the other most common justification for ghostwriting. 58% of respondents surmise that ghostwriting occurs because of "a belief that including co-author information

**Table 5.** Reasons for why ghostwriting may occur.
Themes and supporting examples of write-in responses to the question: "What do you think are the reasons why the names of co-authors on peer review reports may not be provided to the editorial staff?"

| Theme | Example write-in responses |
|---|---|
| Cultural expectations | "A belief that ghostwriting does occur, but everyone accepts that it's just the way it is." |
| | "The belief it has always been like this so why doubt/change the process" |
| | "PIs simply don't think of it because they're used do doing things this way" |
| | "PIs think this practice is okay." |
| Training | "A belief that this is 'how it is done,' and inviting trainees to contribute to reviews is important for their training, but it is not necessarily important for them to get credit for it." |
| | "PI feels while the ECR is being trained in doing the review should not be listed as co-author of the review." |
| "I don't understand it" | "Either as a reviewer or as an editor, I would have no problem with a co-review. I'm not really sure why more people don't do it. They absolutely should." |
| | "I have no idea why this is not common practice." |

DOI: https://doi.org/10.7554/eLife.48425.015

would demonstrate that the PI breached the confidentiality of the manuscript" and 63% surmising that ghostwriting occurs because of "a belief that reviews should only be done by the invited reviewer, and not by, or with assistance from, anyone else." These policies derive from guidelines developed by the Committee on Publication Ethics (COPE): "supervisors who wish to involve their students or junior researchers in peer review must request permission from the editor and abide by the editor's decision" (*COPE Council, 2017*). Yet it seems that, in practice, invited reviewers often do not seek prior permissions, continue to involve ECR co-reviewers, and instead choose to withhold co-reviewer names upon submission. Indeed, 39% of ghostwriters who discussed with the invited reviewer the possibility of including their name cite "journal requires prior approval to share manuscript, which was not obtained" or "journal does not allow ECRs to review" as the reason why their names were withheld. In these cases, adding co-reviewer names to a peer review report is equivalent to admitting that journal policies were disobeyed. Given how frequently ghostwriting occurs based on survey data, and how commonly journal policies are cited as the reason for ghostwriting, it seems that current policies that require invited reviewers to gain permission prior to involving ECRs in peer review are not effective deterrents for ghostwriting. Instead, these policies may have the opposite, if unintended, consequence of preventing invited reviewers from feeling free to add co-

reviewer names upon submission. Journals should acknowledge that peer review is often performed by ECR co-reviewers and remove barriers that prevent ECRs from being named to the editor (*Rodríguez-Bravo et al., 2017*).

The naming of co-reviewers would be facilitated by, for example, a mandatory text-box on the page that reviewers use to submit their review: this page could also contain language asking for co-reviewers be listed. 73% of respondents surmise that ghostwriting occurs because of a lack of such a mechanism. However, when asked to reflect on their own experiences, only 4% of ghostwriters gave this as a reason (though that might be due to a limitation in the design of this question, which did not include this answer in the drop-down menu and instead relied on respondents to write it in). Even with this consideration, these data reveal a difference between the cultural perception of this barrier (73%) and actual experience (4% or more), so any practical solutions (such as adding a text-box) must be accompanied by changes that make it clear that journals expect all co-reviewers to be named.

It is in the best interests of journals to provide mechanisms for ECR co-reviewers to be easily named (*Mehmani, 2019*). If a journal does not know who is reviewing a paper, it cannot be sure that there are no competing interests among the reviewers. Editors may also see an increase in accepted invitations to review once PIs feel free to share the burden of reviewing with their trainees without ethical concerns.

**Table 6.** Reasons given by PI for withholding ECR name.
Themes and supporting examples of write-in responses to the question: "Consider cases where you contributed to a peer review report and you know your name was NOT provided to the editorial staff. When discussing this with your PI, what reason did they give to exclude you as a co-reviewer?".

| Theme | Example write-in response |
|---|---|
| Sin of omission | "They forgot" |
| | "Didn't think of including me; didn't know how to do so" |
| | "He was in a hurry and he couldn't figure out the journal's website" |
| Cultural expectations | "This was not explicitly discussed, but the PI implied this is "common practice" and normal for ECRs to gain experience" |
| | "[PI] said only [they] would be invited to review for such a prestigious journal and "we" need this for future submissions" |
| | "Apparently this duty is part of my job description" |
| | "I was told this is how one gets to train to review papers and grants" |
| A good way to train | "Reviewing papers as [an] ECR is part of the ECR training" |
| | "It's good experience for me." |
| | "It was good for my career to practice." |

DOI: https://doi.org/10.7554/eLife.48425.016

Journals would also benefit from enlarged reviewer pools by including previous ECR co-reviewers and/or consulting existing lists of ECR reviewers (*Burgess, 2018*). Ultimately, journals are responsible for ensuring that all aspects of peer review are fair and ethical. Ghostwriting is neither fair nor ethical, so the evidence for ghostwriting reported here suggests that journals need to update their policies in this area.

### Value judgments about naming co-reviewers

Two hypotheses for why ghostwriting is commonplace were refuted by our data. 82% of respondents disagree that there is no benefit in naming co-reviewers. Co-reviewers, especially ECRs, may value having their name provided to the journal for many reasons, including the ability to be "known" to scientific editors and potential colleagues; the ability to have their work acknowledged by a third party (e.g. Publons) for career advancement; and/or the ability to demonstrate eligibility for residency or visas. ECRs agree more strongly than PIs that there is value in receiving credit or being known to the journal editorial staff as a co-reviewer (*Table 3*), perhaps because invited reviewers are already known as experts in their field. In the words of one write-in response: "PI surprised I would be interested in being acknowledged, and seemed like too much trouble to acknowledge my contribution. There was no box in the online form to declare it." The ambivalence of PIs towards giving due credit for co-reviewers

likely derives from a position of relative privilege. When people of privilege are the only participants in decision-making (for example, on journal editorial boards), they may create policies that fail to consider differing values of less privileged members of the community, like ECRs. We support the growing effort to include more diverse voices in leadership roles in science (such as the "Who's on board" initiative of Future of Research, and the Early-Career Advisory Group at *eLife*).

87% of respondents disagree that ghostwriting occurs because researchers see value in withholding co-reviewer names, perhaps because of a perception that adding co-reviewer names diminishes the review by providing evidence that someone other than the invited reviewer contributed. These data align with results of an ongoing experiment in co-reviewing at the Elsevier journals, in which editors did not rate the co-reviewed reports as low quality, and more than half were considering co-reviewers to serve as invited reviewers on future manuscripts (*Mehmani, 2019*). Research suggests that reviewers who are earlier in their careers may be perceived by editors as better reviewers (*Black et al., 1998*; *Callaham and Tercier, 2007*; *Evans et al., 1993*) and that being closer to bench research, rather than having more experience in reviewing itself, may be a key determinant of this trait (*Stossel, 1985*). Another reason for a perceived value in withholding co-reviewers names is to protect ECRs during a vulnerable time in their careers.

Occasional write-in comments allude to this: "[being named on a peer review report] may give certain ECRs a bad reputation if they review things really harshly" but, on the whole, only 17% of respondents believe protectionism drives ghostwriting. Taken together, respondents find added value in naming co-reviewers and also see no loss of value to peer review when co-reviewer names are added. Finally, the null hypothesis that "a belief that ghostwriting does not occur" was selected least frequently by respondents, demonstrating that ignorance or denial that ghostwriting occurs is rare (*Figure 6*).

## Conclusions

Ghostwriting undermines the integrity of peer review. It is pervasive because many see it as an obligatory feature of peer review training or a necessary delegation of labor. Some don't think to discuss or feel able to discuss authorship on peer review reports. Others are deterred by vague journal policies that do not reflect the status quo – that involving ECRs as co-reviewers is common and considered valuable and ethical. To encourage naming co-reviewers to the editors, journals must clarify their expectations and reporting mechanisms for the participation of ECRs in peer review. These logistical changes should be coupled with an adjustment of cultural expectations for co-review as a training exercise and not exploitation. At a minimum, invited reviewers should discuss with co-reviewers how credit will be given for peer review work. Ideally, they should also ensure that co-review involves feedback so that it is effective training. Changing journal policies and cultural expectations to recognize and value the work of ECRs will benefit the peer review system and all of its constituents.

## Methods

### *Systematic literature review*
Search procedures
The following procedures were used to perform a systematic review of the peer-reviewed literature for any research on the topic of ECR participation in manuscript peer review. These procedures were developed under the guidance of a professional librarian (author SO) and were based on the Preferred Reporting Items for Systematic Reviews and Meta-Analyses (PRISMA) criteria (*PRISMA Group et al., 2009*; *Figure 1—figure supplement 1*). The databases that were searched cover peer-reviewed literature across

the life sciences, public policy and social sciences and were comprised of: PubMed, PsychInfo, Web of Science, and PAIS International. These databases were then searched using the following keyword search strategy: ("early career researcher" OR "graduate student" OR "postdoc" OR "fellow" OR "contingent faculty" OR "adjunct" OR "lecturer" OR "instructor" OR "technician" OR "junior scientist" OR "trainee" OR "lab member" OR "research scientist" OR "postdoctoral fellow" OR "research fellow" OR "teaching fellow" OR "junior researcher" OR "mentee") AND ("peer review" OR "refereeing" OR "invited reviewer" OR "referee" OR "reviewer" OR "co-reviewer" OR "first time reviewers" OR "reviewer training" OR "review partners" OR "contributing author" OR "co-reviewing" OR "reviewing" OR "journal reviewer policy" OR "reviewer guidelines" OR "instructions for reviewers"). These search terms were designed to be broadly inclusive so as to capture any research article with possible relevance to the topic of ECR involvement in manuscript peer review. The resulting collection of 2103 records were imported into the RefWorks 3 bibliographic management database, and duplicate articles were identified and removed using the "Legacy close match" de-duplication filter, resulting in a de-duplicated set of 1952 articles.

Relevance screening
Collected articles underwent two rounds of relevance screening. In the initial round, article titles and abstracts were screened independently by two study authors (JG and GM). Both authors used the same inclusion criteria to sort search results into "relevant," "maybe relevant," and "not relevant" categories. The criteria for article inclusion were: written in English, published in a peer-reviewed journal, mention of ECRs, and mention of peer reviews of manuscripts. Any article that did not meet the inclusion criteria above was excluded as well as database hits for conference proceedings and dissertations.

118 unique articles remained in the "relevant" and/or "maybe relevant" categories at this stage of screening (*Supplementary file 1*). Articles categorized as "relevant" by both initial screeners were selected for full text review (n = 3). Articles categorized as "maybe relevant" by both initial screeners and articles that were differentially categorized as "relevant" vs. "maybe relevant" or "not relevant" by the initial screeners (n = 51) underwent a second round of evaluation by a third, independent screener (author RL) to either confirm categorization as

"relevant" or recategorize as "not relevant" to the topic of ECR participation in the peer review of manuscripts. A resulting list of 36 "relevant" articles underwent a full text reading with specific attention being paid to: research question, motivation for article, method of study including details concerning study participants, relevant results and discussions, discussion of peer review and ECRs, and possible motivations for author bias. Of the articles that were found to be "not relevant" to the topic of ECR participation in the peer review of manuscripts for publication in a journal, most discussed other forms of peer review outside the scope of publishing manuscripts (e.g. students engaging in peer review of each others written work in a classroom setting as a pedagogical exercise).

### Survey of peer review experiences and attitudes

#### Survey design

We designed a survey to evaluate the peer review experiences of researchers with a specific focus on ghostwriting of peer review reports. The survey was verified by the Mount Holyoke Institutional Review Board as Exempt from human subjects research according to 45CFR46.101(b)(2): Anonymous Surveys - No Risk on 08/21/2018. All survey respondents provided their informed consent prior to participating in the survey. The survey comprised 16 questions presented to participants in the following fixed order:

- 6 demographic questions that collected data on their professional status (current institution, field of research, and career stage) and personal information (gender identity, race/ethnicity, and citizenship status in the United States);
- 7 questions that collected data about their experience participating in the peer review of manuscripts for publication in a journal: these questions included 2 questions about their experience with independent reviewing vs. co-reviewing; 4 questions about receiving credit for reviewing activities; and 1 question about whether and how respondents received training in peer review
- 3 questions that collected data about their opinions about co-reviewing and ghostwriting as practices, regardless of whether they had personal experience with these practices: these questions included 1 question about their degree of agreement on a 5-point Likert scale (*Strongly Disagree; Slightly Disagree; No Opinion; Slightly*

*Agree; Strongly Agree*) with 11 statements about the ethics and value of co-reviewing and ghostwriting; 1 question asking their opinion about why ghostwriting as a practice may occur; and 1 exploratory future direction question asking if their opinions would change if the names of peer-reviewers were made publically available ("open peer review").

Throughout the survey, there were many opportunities to provide write-in responses in addition to the multiple choice answers. The full text of the survey can be found in the *Supplementary file 2*.

### Survey distribution, limitations, and future directions

The survey was distributed online through channels available to the nonprofit organization Future of Research including via blog posts (*McDowell and Lijek, 2018*), email lists, social media, and word-of-mouth through colleagues. The main survey data collection effort was from August 23 to September 23, 2018. The survey had gathered 498 responses at the time of data analysis.

We recognize that conclusions drawn from any survey data are limited by the size and sample of the population that is captured by the survey. We sought to address this limitation first by collecting as large and geographically and institutionally diverse of a population of ECRs as possible within the month-long timeframe we set for data collection. Secondly, we wished to preemptively address the concern that our survey distribution efforts were inherently biased towards those with strong opinions on the subject and/or those who self-select to receive communication from Future of Research (e.g. listservs, Twitter followers). We therefore attempted to create a "negative control" comparison group of participants who received our survey from channels independent of Future of Research. We created a separate survey form asking identical questions and personally asked 25+ PIs known to the authors, as well as organizational collectives of PIs, to distribute this survey link to their own networks (e.g. lab members, departments). Both surveys were live during the same month-long time period; however, the PI-distributed survey gathered only 12 responses and so was not sufficient to be used in the analyses presented here. Since the goal of the second, PI-distributed survey was to be independently distributed outside of our efforts, we

are not able to determine whether it garnered so few responses because of a lack of genuine distribution or because the populations it reached were not motivated to participate in the survey. Therefore any conclusions drawn from this study reflect the 498 experiences and perspectives of those individuals so moved to participate in the survey distributed by Future of Research and our results should be considered in this context. One possible future direction for this study is to reopen the survey in conjunction with publication of this manuscript in an effort to broaden and diversify the sampled population, to compare subsequent rounds of responses to our initial 498 responses, and to improve clarity on survey questions (Appendix 2).

## Survey data analysis

Survey data were analyzed using Microsoft Excel, Version 16 and IBM SPSS Statistics for Macintosh, Version 25 (SPSS Inc, Chicago, IL, USA). Whenever statistical analyses were used, the exact tests and p values are reported in the appropriate figure legend and/or results text. A p value of less than 0.05 was considered significant.

## Acknowledgements

The authors thank Dr Jessica Polka, Dr Adriana Bankston, Dr Carrie Niziolek and Dr Chris Pickett for their constructive comments on this manuscript. GSMD thanks the Open Philanthropy Project for providing support for his role at the Future of Research.

**Gary S McDowell** is at the Future of Research, Inc, Abington, United States
garymcdow@gmail.com
 https://orcid.org/0000-0002-9470-3799

**John D Knutsen** is in the Department of Psychology, Harvard University, Cambridge, United States

**June M Graham** is in the Department of Biological Sciences, Mount Holyoke College, South Hadley, United States

**Sarah K Oelker** is in the Division of Research & Instructional Support, Mount Holyoke College, South Hadley, United States
 https://orcid.org/0000-0001-6655-7184

**Rebeccah S Lijek** is in the Department of Biological Sciences, Mount Holyoke College, South Hadley, United States
rlijek@mtholyoke.edu
 https://orcid.org/0000-0003-2474-6870

**Author contributions:** Gary S McDowell, Conceptualization, Supervision, Visualization, Methodology, Writing—original draft, Writing—review and editing; John D Knutsen, Formal analysis, Methodology, Writing—review and editing; June M Graham, Sarah K Oelker, Data curation, Investigation, Methodology, Writing—original draft; Rebeccah S Lijek, Conceptualization, Formal analysis, Supervision, Investigation, Methodology, Writing—original draft, Project administration, Writing—review and editing

**Competing interests:** Gary S McDowell: Received financial compensation as the Executive Director of the 501(c)3 non-profit Future of Research, Inc. Rebeccah S Lijek: Former volunteer member of the Board of Directors of Future of Research, Inc. The other authors declare that no competing interests exist.

**Ethics:** Human subjects: The survey was verified by the Mount Holyoke College Institutional Review Board as Exempt from human subjects research according to 45CFR46.101(b)(2): Anonymous Surveys - No Risk on 08/21/2018. Prior to participating in the survey, all respondents were presented with the following statement and then selected a checkbox to provide their informed consent: "By choosing to submit answers to this survey, you thereby provide your informed consent to voluntarily share your experiences and opinions with the researchers, who intend to publish a summary of the results of the survey but not the raw data with participants' individual demographic information." See Supplementary Materials for full Statements of Disclosure, Ethics and Informed Consent.

### Funding

| Funder | Grant reference number | Author |
|---|---|---|
| Open Philanthropy Project | Future of Research - General Support | Gary S McDowell |

The funders had no role in study design, data collection and interpretation, or the decision to submit the work for publication.

**Decision letter and Author response**
Decision letter https://doi.org/10.7554/eLife.48425.024
Author response https://doi.org/10.7554/eLife.48425.025

## Additional files

### Supplementary files

• Supplementary file 1. Results of relevance screening for literature review.
DOI: https://doi.org/10.7554/eLife.48425.017

• Supplementary file 2. Text of The Role of Early Career Researchers in Peer Review – Survey.
DOI: https://doi.org/10.7554/eLife.48425.018

• Transparent reporting form DOI: https://doi.org/10.7554/eLife.48425.019

## Data availability

Literature review results are shared in supplementary materials; De-identified source data for Figures 1 and 5 have been provided in response to editorial request. Raw data from the survey are not shared to protect respondents' confidentiality.

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

## Appendix 1

DOI: https://doi.org/10.7554/eLife.48425.020

### Literature on ECR involvement in peer review as a training exercise and examples of formalized training in peer review

Many of the articles uncovered by our systematic literature review on the topic of ECR involvement in peer review of manuscripts that *did not* address our desired topic of ghostwriting instead discussed ECR involvement in peer review of manuscripts as a training exercise for the ECR. Here, we summarize their findings. We also give some examples of formalized training in peer review.

### Peer review training programs with positive outcomes

Trends from the literature indicate several components of a successful peer review training program and where such programs are currently in process. Authors studying the peer review training process for new reviewers tend to conclude that students of peer review learn best by participating themselves in the review process while receiving feedback from more senior reviewers. Successful training programs, in which the reviewers expressed that they had benefited from the training or were evaluated and determined to have benefited from the training, tended to include participation in several rounds of review followed by feedback and revising. Additionally, successful programs often expanded feedback by directing the expert reviewers to report on the same manuscripts as the trainees. This method provided trainees with specific written feedback as well as a pertinent example report to reference. Authors of successful program studies and authors with general policy recommendations for peer review training strategies conclude that a hands-on, iterative process of peer review training with regular and specific feedback are components which positively benefit the peer review trainee (*Castelló et al., 2017*; *Doran et al., 2014*).

### Peer review training programs without positive outcomes

However, one reviewer training study in the pool generated negative results: Houry, Green and Callaham found that after a period of training involving mentorship from an expert reviewer to a new reviewer, no differences in mean reviewer quality scores between the mentored and unmentored groups was found (*Houry et al., 2012*). The study goes on to conclude that this similarity in group quality scores is dependent on the mentoring relationship: the expert reviewer mentors were not given any training on how to offer feedback to the trainees. This aspect of the program was deliberately constructed in an attempt to model a training program in which there would be minimal stress on the expert mentors. However, it appears that this ultimately led to an inconsequential training program. Accounting for this information, Houry et al. conclude that a mentorship program should include training and guidelines for mentor-mentee communications which allow for regular feedback from expert to trainee.

### Institutions where peer review training takes place

Sources mentioned two primary institutions where training in peer review may take place. Training institutions were identified, with undergraduate education as the first opportunity for training, followed by graduate programs. Journals were the other institutions identified. The majority of training programs represented in the sources take place within a journal setting. Journal programs tend to include ECRs, such as undergraduate or graduate students, joining the editorial board for a set time period or acting as a reviewer (*Castelló et al., 2017*; *Doran et al., 2014*; *Harrison, 2009*; *Houry et al., 2012*; *Navalta and Lyons, 2010*; *Patterson and Schekman, 2018*; *Picciotto, 2018*).

Training may also take place within undergraduate or graduate institutions. We found one study about training in peer review in an undergraduate setting. Despite the lack of associated literature, Riehle and Hensley determined that undergraduate students are interested in learning about the peer review process (*Riehle and Hensley, 2017*). In a training study, the Calibrated Peer Review system was employed in two undergraduate classes to facilitate students to peer review the work of their classmates while minimizing the extra workload such an exercise might otherwise entail for the professor (*Prichard, 2005*). The participating students were in two separate courses, an introductory neuroscience course and a more advanced neuroscience course. Students in the advanced class did not perform better on peer review exercises than the introductory students, suggesting that until that point, advanced students had not been exposed productively to peer review practice. Authors deemed this to be a successful method for exposing undergraduate students to the peer review process while requiring a realistic time commitment from the course instructor.

While no papers were found detailing the effectiveness of peer review training within graduate institutions, several sources did indicate graduate student perceptions about their program peer review training. In a study including psychology masters and PhD students, a large proportion of participants indicated that their education had lacked in providing information on the peer review and revision process as well as information about how to practice review (*Doran et al., 2014*). These students indicate that this was a negative aspect of their programs, saying more opportunities for peer review practice should be made available. Authors of this article do indicate that this information may not be generalizable to graduate students as a group because the participating students were found when they pursued a journal review program, something students with adequate peer review education may not be likely to do.

### Self facilitated training

Merry et al. provides a list of recommendations for ECRs to facilitate their own training of peer review (*Merry et al., 2017*). Authors advise working with the mentoring faculty as well as contacting journals directly to seek out peer review opportunities. If mentors are able to give consistent feedback regarding the trainees peer review, it seems that this could be a positive environment in which to learn the skill.

### Roadblocks to positive outcomes in training programs

Papers discussing journal training programs cite feasibility as the largest roadblock to success (*Castelló et al., 2017*; *Houry et al., 2012*). As discussed, it is recommended that peer review training programs for ECRs feature a system which provides regular, specific feedback from expert reviewers. Such programs require high levels of labor, involving organization and time commitment from program leaders and expert reviewers. This is a significant investment for a business which may be in conflict with the desire to maximize journal profit. One possible solution presented involves student-run journals hiring increased numbers of student reviewers and editors so experience may be gained in the field (*Doran et al., 2014*; *Patterson and Schekman, 2018*). However, this solution does not address the recommendation for expert reviewers to provide feedback.

### Examples of formalized training in peer review

Some graduate programs, scientific societies and journals already provide materials for training in peer review. For example, the American Chemical Society provides a free course designed by editors, researchers and publication staff (acsreviewerlab.org). GENETICS, a journal published by the Genetics Society of America, provides a peer review training program with virtual training sessions for ECR members (genetics-gsa.org/careers/training_program.shtml). Nature also provides an online course on peer review (masterclasses.nature.com/courses/205). Publons (which provides a mechanism for recording and crediting peer review

activity) has an online peer review academy (publons.com/community/academy/). The Journal of Young Investigators provides training at the undergraduate level (jyi.org).

We also found a number of examples of courses and classes that train graduate students in peer review. These include: a class run by Needhi Bhalla at the University of California Santa Cruz; Class 230 in the PhD Program in Biological and Biomedical Sciences at Harvard; a class on critiquing papers at University of California San Diego; a class in the graduate program in Systems Biology at Harvard Medical School; and a requirement to review a paper as a final exam recently introduced at the Graduate Field of Biochemistry, Molecular, and Cell Biology at Cornell University. The University of Texas Southwestern is experimenting with an advanced course combining peer review, literature review, debate and commentary communication and lay-audience oriented writing. A number of journal clubs also review preprints and send comments to authors (*Avasthi et al., 2018*).

## Appendix 2

DOI: https://doi.org/10.7554/eLife.48425.020

### Future directions for survey questions

#### Opinions on open peer review reports and public naming of reviewers

The final (16th) question of the survey provided respondents with an open comment box in which to reflect on whether any of their opinions would change if either the *contents* of peer review reports or the *names of reviewers* were openly published to journal readership. This question is a stark contrast to the rest of the survey questions, all of which instead only ask about naming co-reviewers to the journal staff and editors but not openly to the public readership. We included this question as a test balloon for future surveys that might focus on open peer review vs. traditional models of closed peer review. Due to its tangential relationship to the goals of the current study on ghostwriting and due to the question's open-ended, write-in nature, we did not perform a systematic analysis of responses as with the other questions. A qualitative summary of responses is below.

61% of respondents chose to write a response to this question, and of these approximately one third reported that no, their opinions would not change if the peer review reports nor names of reviewers were openly shared with the public readership (it should be noted that these comments mostly, but not necessarily, endorsed the specific open peer review features suggested in the question). The remaining respondents expressed a variety of concerns, mostly surrounding the loss of anonymity of reviewers rather than what appeared to be the less controversial concept of publishing the contents of peer review reports. Respondents' hesitations about anonymity often centered on the effect that naming ECRs and URMs might have on these vulnerable populations. These responses reflect other conversations about open peer review (*Polka et al., 2018*; *Ross-Hellauer, 2017*; *Ross-Hellauer et al., 2017*; *Tennant, 2017*) and in the context of recent data about referee behavior in open peer review (*Bravo et al., 2019*) warrant further analysis beyond the scope of this manuscript.

#### Improving clarity in survey questions

One survey question that would benefit from disambiguation in future iterations of the survey is: "Agree/Disagree: When a journal invites a PI to review, that is equivalent to the journal inviting anyone in that PI's research group with relevant expertise to contribute to the review." It was brought to our attention by respondents' emails and write-in comments during the survey response period that there was confusion about this statement. We had hypothesized that we might find agreement with the statement; however, we found a substantial amount of disagreement which may be due in part to the various ways the question may have been interpreted. Our intention was to determine if respondents agreed that, in practice, it could be reasonable for all engaged in journal publication and peer review to expect that an invited reviewer would have various motivations to share a manuscript with the relevant expertise in their research group, particularly in cases where a postdoc is likely more familiar with the literature or experimental techniques than a Principal Investigator overseeing a number of projects. However it became apparent that this question was quite open to various interpretations as described in write-in comments received on the question, including:

- Does the respondent believe this *should* be the case?
- Does the respondent believe this is what is *actually happening*?
- Does the respondent consider that journals have this intent when they invite reviewers?

This therefore renders interpretation of responses to this question difficult, and so we chose not to draw conclusions from this particular result. We aim to clarify this question should there be future iterations of this survey.

Another survey question that would benefit from future disambiguation draws from respondents' ability to select multiple responses for the question "When you were not the

invited reviewer, what was the extent of your involvement in the peer review process? Please select all that apply to your entire peer review experience (e.g. across multiple manuscripts)." and indeed the inability to discern whether respondents supplying only one answer were selecting that response because that comprised the totality of their experiences, or because they selected the most common experience. In comparing responses to this question with other questions, it may be that there are analyses that are affected by the assumption that it is possible to apply one response of many to a response to another question. We attempted to preemptively disambiguate responses by asking whether respondents had "ever" experienced certain things in subsequent questions.

