## [Decision Letter]

Thank you for submitting your article "Co-reviewing and ghostwriting by early career researchers in the peer review of manuscripts" for consideration by eLife. Your article has been reviewed by 3 peer reviewers, and the evaluation has been overseen by myself (Emma Pewsey) as Reviewing Editor and Peter Rodgers, the eLife Features Editor. The following individuals involved in review of your submission have agreed to reveal their identity: Blanca Rodríguez-Bravo (Reviewer #1); Sarvenaz Sarabipour (Reviewer #2); Francisco Grimaldo (Reviewer #3).

The reviewers have discussed the reviews with one another and I have drafted this decision to help you prepare a revised submission. We hope you will be able to submit the revised version within two months.

-----

SUMMARY

By surveying 489 researchers, the authors show that co-reviewing performed by ECRs when their PI is invited to review manuscripts is the norm and that ghostwriting (i.e. ECRs reviewing papers in the name of someone else, often their PI) is common despite being deemed unethical by the majority of survey respondents. This is a timely study, and the manuscript is solid: the objectives of the study are clear, the methodology is explained conveniently with the help of the supplementary materials that are provided, and the discussion is broad and relevant.

ESSENTIAL REVISIONS

1. Please state if the responses to any of the questions have been analysed by gender, field (life sciences etc.), citizenship and race/ethnicity/URM of the participants. If so, please discuss any differences that can be observed.

2. Figure 3: From the survey data, PIs and journal clubs are the two most important sources of reviewer training. It is therefore worth noting in the manuscript that PIs should take this career development training and mentoring more seriously. You could potentially also mention that PREReview (https://www.authorea.com/inst/14743-prereview) and preprint journal club reviews can complement PI mentorship and help train and prepare ECRs for independent and co-reviewer roles.

3. Lines 228-231: "In 3 of 11 statements, ECRs felt significantly more strongly than PIs but still shared the same valance ". Please clarify whether any significant differences were found in the options of ECRs and PIs for the other 8 statements.

4. Line 232: Please explain why you chose to include a 'no opinion' option when asking participants how strongly they agreed with the statements. Is this a necessary option for future surveys of this kind?

5. In the discussion section, please discuss whether there is a way to incentivize journals to provide mechanisms for PIs to acknowledge the contributions of ECRs to reviews.

6. The discussion and conclusions advocate for a cultural shift, among others, but it would be nice to also discuss the change of mind that affects ECRs when they become PIs in such a hypercompetitive environment. Are there new incentives that could be introduced to encourage PIs to name their ECR co-reviewers?

7. Figure 7: Please ensure that each reason given in this graph is addressed in the "Discussion" of the manuscript and offer potential practical solutions for each to the community. For instance, in response to the statement that "73% of respondents reported that they had not discussed being credited for co-reviewing with their PIs" (line 269), you could mentions that better mentoring practices help overcome such miscommunication.

OPTIONAL SUGGESTIONS

a. You may wish to consult the results of the project "Early Career Researchers the Harbingers of Change?" (http://ciber-research.eu/harbingers.html), in particular the publication focused on peer review: Blanca Rodríguez-Bravo, David Nicholas, Eti Herman, Chérifa Boukacem-Zeghmouri, Anthony Watkinson, Jie Xu, Abdullah Abrizah, Marzena Świgoń Peer review: The experience and views of early career researchers Learned Publishing, 30(4), pp.269-277.

One of the recommendations of the article is that journals should acknowledge that peer review is often performed by co-reviewers, and facilitate it for training purposes, since in this way the principal investigators will have no excuse for not giving credit to their collaborators.

b. You may wish to discuss ways of reducing the need for co-reviewing. For example, starting a database of ECR reviewers can help train potential reviewers and ensure a productive peer review process. Recommending this list in the manuscript to journals/associate editors may help to practically address ECR ghost reviewing by contacting ECRs directly and acknowledging their expertise. An example of such a list is: https://ecrlife.org/2018/03/02/boosting-ecr-involvement-in-peer-review/

---

## [Author Response]

We repeat the reviewers’ points here in italic, and include our replies point by point, as well as a description of the changes made, in Roman.

**ESSENTIAL REVISIONS**

1. Please state if the responses to any of the questions have been analysed by gender, field (life sciences etc.), citizenship and race/ethnicity/URM of the participants. If so, please discuss any differences that can be observed.

In addition to the analysis by career stage described in the paper (ECRs vs PIs), we also performed preliminary analyses by gender, field, citizenship and race/ethnicity of the participants because we were also very interested in these comparisons. However, there were no striking conclusions to be drawn from these demographic analyses: differences were not statistically significant nor of great magnitude and some demographic subsets (eg. U.S. domestic people of color in the life sciences) were underpowered with small n values. To report these analyses thoroughly without cherry picking would significantly increase the length of an already long manuscript. Early drafts of our preprint included some of these descriptions of demographic data and were flagged by colleagues as tangential and distracting. We are considering how to share these negative data more completely, e.g. in a preprint or blogpost, and believe they are beyond the scope of this manuscript. We hope the reviewers find this reasonable and acceptable.

2. Figure 3: From the survey data, PIs and journal clubs are the two most important sources of reviewer training. It is therefore worth noting in the manuscript that PIs should take this career development training and mentoring more seriously. You could potentially also mention that PREReview (www.authorea.com/inst/14743-prereview) and preprint journal club reviews can complement PI mentorship and help train and prepare ECRs for independent and co-reviewer roles.

We completely agree with the reviewers’ point that PIs must take this training responsibility more seriously - thank you! We also concur that gaining review experience through preprint (or regular) journals clubs is an important complement to PIs’ mentorship. Note that preprint journal clubs were already mentioned briefly in Supplementary File 3: Examples of Formalized Training in Peer Review, citing Avasthi et al. 2018. This section also references a template for how to integrate preprint journal clubs into the graduate classroom provided by Dr. Needhi Bhalla at the University of California Santa Cruz that is included in our preprint.

We added new language on line 297 to better highlight our recommendations for training in peer review, and in particular the role that PIs and journal clubs serve:

“Given their key role as the main source of training in peer review, it is important that PIs make deliberate efforts to teach their trainees this skill and provide feedback on ECRs’ work. Mentorship from PIs may be complemented with, but not replaced by, journal clubs where ECRs gain experience in reviewing published manuscripts or preprints (e.g. Avasthi et al. 2018 and PREReview https://www.authorea.com/inst/14743-prereview). Since these ad hoc reviewing experiences with PIs and journal clubs may vary in their availability and quality, all ECRs should be offered standardized, evidence-based training in peer review. For example, peer review courses that are compulsory and ubiquitous in graduate schools would ensure that all PhD-holders are appropriately trained to perform constructive peer review.”

3. Lines 228-231: "In 3 of 11 statements, ECRs felt significantly more strongly than PIs but still shared the same valance ". Please clarify whether any significant differences were found in the options of ECRs and PIs for the other 8 statements.

We added the following sentence immediately after the one quoted above at line 174: “For the remaining 8 statements, ECRs and PIs did not significantly differ in their opinions.”

4. Line 232: Please explain why you chose to include a 'no opinion' option when asking participants how strongly they agreed with the statements. Is this a necessary option for future surveys of this kind?

A ‘no opinion’ option is not necessary for future surveys, but it is our preference for this and future surveys. The ‘no opinion’ option with the Likert scale provides respondents with an opportunity to demonstrate their true neutral opinion, rather than forcing them to agree or disagree. This creates cleaner, more interpretable data (see references below on evidenced-based survey design). If a respondent has no opinion on a topic but is not offered a “no opinion” option in the survey, they would either have to a) choose an inauthentic opinion from the survey choices or b) leave the question blank. Choosing an opinion even when one has none (a forced choice) would introduce noise into the survey results. Leaving the question blank would create a missing data point. A missing data point would not be sufficient for us to interpret it as “no opinion” since there are other reasons why a question might be skipped (e.g. accidental oversight, or “I don’t understand”).

We included a “no opinion” option in our survey to be inclusive of the reality that respondents may feel neutrally about certain aspects of co-reviewing and ghostwriting. We wanted our dataset to be able to capture that indifference, should it exist, as much as we wanted to know whether respondents’ agreed or disagreed. For example, we hypothesized that PIs’ indifference about the value of naming co-reviewers might cause them to omit co-reviewer names. Evidence in support of that hypothesis is described at line 410 of the manuscript:

“ECRs agree more strongly than PIs that there is value in receiving credit or being known to the journal editorial staff as a co-reviewer (Table 3), perhaps because invited reviewers are already known as experts in their field. In the words of one write-in response: “PI surprised I would be interested in being acknowledged, and seemed like too much trouble to acknowledge my contribution. There was no box in the online form to declare it.” PIs’ ambivalence for due credit for co-reviewers likely derives from a position of relative privilege. When people of privilege are the only participants in decision-making (e.g. on journal editorial boards), they may create policies that fail to consider differing values of less privileged members of the community, like ECRs.”

Examples of research on the utility of the midpoint “no opinion” option:

· Chyung, S. Y., Roberts, K., Swanson, I., & Hankinson, A. Evidence-based survey design: The use of a midpoint on the Likert scale. Performance Improvement Journal. 2017; 56(10).

· Kulas JT, Stachowski AA. Middle category endorsement in odd-numbered Likert response scales: associated item characteristics, cognitive demands, and preferred meanings. J. Res. Pers. 2009;43:489–493.

· Nadler, J. T., Weston, R., & Voyles, E. C. Stuck in the middle: The use and interpretation of mid-points in items on questionnaires. The Journal of General Psychology. 2015; 142(2), 71-89.

5. In the discussion section, please discuss whether there is a way to incentivize journals to provide mechanisms for PIs to acknowledge the contributions of ECRs to reviews.

We strongly agree with the reviewers that systematic change will require thoughtful reconsideration of the current incentives for journals, PIs, etc. To this effect, we have added the following text at line 393 of the manuscript:

“It is in journals’ best interest to provide mechanisms for ECR co-reviewers to be easily named, as described recently by an Elsevier editor (Mehmani, 2019). Editors take care to assign appropriate reviewers that lack conflicts of interest, an effort that is stymied by ghostwriting. Editors may also see an increase in accepted invitations to review once PIs feel free to share the burden of reviewing with their trainees without ethical concerns. Journals would also benefit from enlarged reviewer pools by including previous ECR co-reviewers and/or consulting existing lists of ECR reviewers (https://ecrlife.org/2018/03/02/boosting-ecr-involvement-in-peer-review). Ultimately, the burden of proof for ethical behavior falls on journals who are the managers of the peer review process. The status quo where ghostwriting abounds and is widely considered unethical is a strong disincentive for maintaining current policies.”

More generally, we note that reviewers’ comments #5-7 and optional comment #2 all make requests to add language about how we might change the system. We are hesitant to include too many opinions about how to incentivize policy and behavioral changes in this manuscript in an effort to separate the unbiased reporting of data (in this manuscript) from personal opinions. We plan to share our recommendations for policy changes in a companion opinion article, currently in preparation, that draws on the data reported here. We are happy to provide these reviewers with a draft of that opinion manuscript should they wish to see it.

6. The discussion and conclusions advocate for a cultural shift, among others, but it would be nice to also discuss the change of mind that affects ECRs when they become PIs in such a hypercompetitive environment. Are there new incentives that could be introduced to encourage PIs to name their ECR co-reviewers?

We don’t interpret these data to mean that ECRs change their minds on these issues when they become PIs. Instead, it seems as though PIs simply re-create their own experiences because that is “how it is done” (see Table 5). So if someone was not named to the journal when they were an ECR co-reviewer, then they do not name their own trainees as co-reviewers once they become a PI, even if they think that is how it *should* be done. Junior/pre-tenure PIs in particular may hesitate to do anything differently for fear it might reflect poorly upon them during their own tenure evaluation.

In our opinion, the best way to incentivize PIs to do something is to make it a requirement of grant funding or tenure. In general, mentorship activities are not well accounted for in these evaluation processes and ought to be. For example, perhaps PIs who pay ECR salaries from grant budgets should demonstrate in grant progress reports how those ECRs are being trained, which might include a list of which trainees were listed as co-reviewers by journals or Publons.

We added this text to line 333 of the manuscript: “PIs could be further incentivized to name ECR co-reviewers by accounting for this practice in PIs’ evaluation for grant funding or tenure. For example, perhaps PIs who pay trainees’ salaries from grant budgets should demonstrate in grant progress reports how those ECRs are being trained, which might include a list of which trainees were listed as co-reviewers.”

Journals have the power to make naming co-reviewers so easy and obvious during the peer review report submission process that it may not require further incentivization for the PI. In other words, we don’t think that a carrot needs to be added to the system, but rather the sticks (barriers to good behavior) need to be removed. If journals make naming co-reviewers a clear, standard expectation of all invited reviewers, then it seems unlikely that a PI would actively avoid naming a co-reviewer. In those rare cases, perhaps journals could include a whistleblower hotline for a ghostwriter to alert the journal in cases where names were intentionally and maliciously withheld. A particularly strong disincentive to hiding ECR contributions is that this has been used as a pretext to dismiss faculty where ECRs have been shown confidential grant applications to assist with reviews (e.g. https://www.bloomberg.com/news/features/2019-06-13/the-u-s-is-purging-chinese-americans-from-top-cancer-research).

7. Figure 7: Please ensure that each reason given in this graph is addressed in the "Discussion" of the manuscript and offer potential practical solutions for each to the community.For instance, in response to the statement that "73% of respondents reported that they had not discussed being credited for co-reviewing with their PIs" (line 269), you could mentions that better mentoring practices help overcome such miscommunication.

We have made sure that all of the possible responses in (previously Figure 7) now Figure 6 are discussed with possible solutions (noted by “→” below) on the following pages:

No practical mechanism pg 17

“A lack of a mechanism (such as a text box), with language demonstrating expectations that co-reviewers be listed) to include this information in the peer review report submission process.”

→ add textbox along with clear language upon submission of the peer review report that invited reviewers are expected to name co-reviewers or acknowledge the review was only performed by the invited reviewer

Prohibitive journal policies pg 16

“A belief that reviews should only be done by the invited reviewer, and not by, or with assistance from anyone else.”

“A belief that including co-author information would demonstrate that the PI breached the confidentiality of the manuscript.“

→ remove or reform current prior permissions/confidentiality policies

Co-reviewers don’t deserve authorship: pg. 15

“A belief that only the invited reviewer deserves authorship, even when others contribute ideas and/or text to the review report.”

“A belief that there is no strong ethical reason to add co-authors names.”

→ journals and PIs should realize that respondents think that co-reviewers do deserve authorship and adjust their policies/behaviors to reflect this by routinely naming all co-reviewers

Power dynamics pg 15

“Some ECRs may not be comfortable asking for co-authorship.”

→ PIs should initiate these discussions

Protectionism pg 19

“Some co-reviewers want to be able to write critical reviews anonymously.”

“A belief that keeping ECR names off of peer review protects ECRs during a vulnerable time in their career.”

→ Not a major driver of ghostwriting, no changes needed

Denialism and/or ignorance pg. Pg 19

“A belief that ghostwriting does not occur: PIs are the only people that contribute to peer review reports.”

“A belief that ghostwriting does not occur: everyone always provides the names of all contributing authors to the editorial office.”

→ Not a major driver of ghostwriting, no changes needed

The data referenced by the reviewers’ comment above from line 269 (in first submission, now line 204 in main manuscript without embedded figures) is from Table 4 and we have added the point about mentorship to the discussion of that table at line 332: “If ghostwriting arises from PIs and ECRs simply not thinking to include co-reviewer names vs. intentional withholding of names, then building awareness should encourage more conversation about this issue *and better mentoring practices will help overcome such miscommunication*.”

**OPTIONAL SUGGESTIONS**

a. You may wish to consult the results of the project "Early Career Researchers the Harbingers of Change?" (http://ciber-research.eu/harbingers.html), in particular the publication focused on peer review: Blanca Rodríguez-Bravo, David Nicholas, Eti Herman, Chérifa Boukacem-Zeghmouri, Anthony Watkinson, Jie Xu, Abdullah Abrizah, Marzena Świgoń Peer review: The experience and views of early career researchers Learned Publishing, 30(4), pp.269-277. One of the recommendations of the article is that journals should acknowledge that peer review is often performed by co-reviewers, and facilitate it for training purposes, since in this way the principal investigators will have no excuse for not giving credit to their collaborators.

Thank you, we have added that citation and the following text on line 377 of the manuscript: “Journals should acknowledge that peer review is often performed by ECR co-reviewers and remove barriers that prevent ECRs from being named to the editor (Rodríguez-Bravo et al., 2017).”

b. You may wish to discuss ways of reducing the need for co-reviewing. For example, starting a database of ECR reviewers can help train potential reviewers and ensure a productive peer review process. Recommending this list in the manuscript to journals/associate editors may help to practically address ECR ghost reviewing by contacting ECRs directly and acknowledging their expertise. An example of such a list is: https://ecrlife.org/2018/03/02/boosting-ecr-involvement-in-peer-review/

We have incorporated this into the response to point #5 above, now on line 397 of the manuscript. “Journals would also benefit from enlarged reviewer pools by including previous ECR co-reviewers and/or consulting existing lists of ECR reviewers (https://ecrlife.org/2018/03/02/boosting-ecr-involvement-in-peer-review).”

**Additional changes made to manuscript in response to editorial request**

(Added on September 22, 2019: Please see bottom of letter for list of additional changes to the manuscript that were requested by the editors. All of these changes have been made and the line numbers in the letter to reviewers’ have been updated accordingly.)

AUTHOR CONTRIBUTIONS

- author contributions text has been removed from the manuscript. There is a still a short section about financial disclosures.

BOXES

- converted all Boxes in into tables

- removed the top title bar from each

- reordered tables to reflect the order of first mention in manuscript text

FIGURES

- Combined old Figures 4&5 into new Figure 4A and 4B

- Added ratios to figure legend for new Figure 4

- Renumbered remaining figures

- Added requested text/ratios to Figure 1 legend

SUPPLEMENTAL FILES

- new Supplemental Files 1-3 created as requested

- formerly Supplemental Figure 1 (flow chart of literature review) changed to Figure 1 Supplement, as requested

SOURCE DATA

- deidentified source data provided for Figure 1 (demographics) and Figure 5 (formerly Figure 6 as requested, opinions)